# Topology-Imbalance Learning
# for Semi-Supervised Node Classification

**Deli Chen**[1,2], **Yankai Lin**[1], **Guangxiang Zhao**[2], **Xuancheng Ren**[2],
**Peng Li**[1], **Jie Zhou**[1], **Xu Sun**[2]
[1]Pattern Recognition Center, WeChat AI, Tencent Inc., China
[2]MOE Key Lab of Computational Linguistics, School of EECS, Peking University
`{delichen, yankailin, patrickpli,withtomzhou}@tencent.com`
`{zhaoguangxiang,renxc,xusun}@pku.edu.cn`

## Abstract

The class imbalance problem, as an important issue in learning node representations, has drawn increasing attention from the community. Although the imbalance considered by existing studies roots from the unequal quantity of labeled examples in different classes (*quantity imbalance*), we argue that graph data expose a unique source of imbalance from the asymmetric topological properties of the labeled nodes, i.e., labeled nodes are not equal in terms of their structural role in the graph (*topology imbalance*). In this work, we first probe the previously unknown topology-imbalance issue, including its characteristics, causes, and threats to semi-supervised node classification learning. We then provide a unified view to jointly analyzing the quantity- and topology- imbalance issues by considering the node influence shift phenomenon with the Label Propagation algorithm. In light of our analysis, we devise an influence conflict detection–based metric Totoro to measure the degree of graph topology imbalance and propose a model-agnostic method ReNode to address the topology-imbalance issue by re-weighting the influence of labeled nodes adaptively based on their relative positions to class boundaries. Systematic experiments demonstrate the effectiveness and generalizability of our method in relieving topology-imbalance issue and promoting semi-supervised node classification. The further analysis unveils varied sensitivity of different graph neural networks (GNNs) to topology imbalance, which may serve as a new perspective in evaluating GNN architectures.[1]

## 1 Introduction

Graph is a widely-used data structure [51], where the nodes are connected to each other through natural or handcrafted edges. Similar to other data structures, the representation learning for node classification faces the challenge of quantity-imbalance issue, where the labeling size varies among classes and the decision boundaries of trained classifiers are mainly decided by the majority classes [46]. There have been a series of studies [35, 11, 49] handling the Quantity-Imbalance Node Representation Learning (short as QINL). However, different with other data structures, graph-structured data suffers from another aspect of the imbalance problem: the imbalance caused by the asymmetric and uneven topology of labeled nodes, where the decision boundaries are driven by the labeled nodes close to the topological class boundaries (left of Figure 1) thus interfering with the model learning.

**Present Work.** For the first time, we recognize the **Topology-Imbalance Node Representation Learning** (short as TINL) as a graph-specific imbalance learning topic, which mainly focus on the

---

[1]The code is available at https://github.com/victorchen96/ReNode.

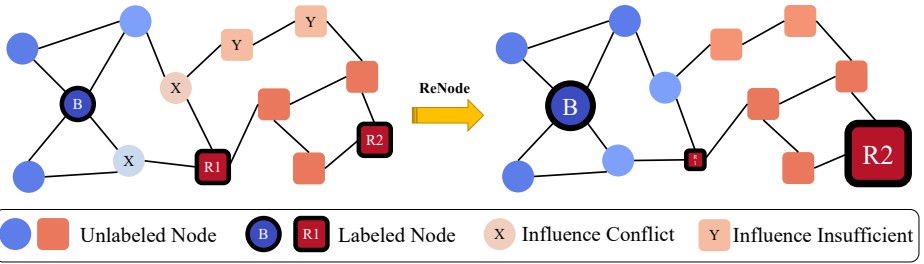

Figure 1: Schematic diagram of the topology-imbalance issue in node representation learning. The color and the hue denote the type and the intensity of each node's received influence from the labeled nodes, respectively. The left shows that nodes close to the boundary have the risk of information conflict and nodes far away from labeled nodes have the risk of information insufficient. The right shows that our method can decrease the training weights of labeled nodes (R1) close to the class boundary and increase the weights of labeled nodes (B and R2) close to the class centers, thus relieving the topology-imbalance issue.

decision boundaries shift phenomena driven by the topology imbalance in graph and is an essential component for node imbalance learning. Comparing with the well-explored QINL that studies the imbalance caused by the numbers of labeled nodes, TINL explores the imbalance caused by the positions of labeled nodes and owns the following characteristics:

- **Ubiquity**: Due to the complex connections of the graph nodes, the topology structure of nodes in different categories is naturally asymmetric, which makes TINL an essential characteristic in node representation learning. Hence, it is difficult to construct a completely symmetric labeling set even with an abundant annotation budget.

- **Perniciousness**: The influence from labeled nodes decays with the topology distance [3]. The asymmetric topology of labeled nodes in different classes and the uneven distribution of labeled nodes in the same class will cause the influence conflict and influence insufficient problems (left of Figure 1) respectively, resulting in a shift of decision boundaries.

- **Orthogonality**: Quantity-imbalance studies [49, 8, 5] usually treat the labeled nodes of the same class as a whole and devise solutions based on the total numbers of each class, while TINL explores the influence of the unique position of each labeled node on decision boundaries. Thus, TINL is independent of QINL in terms of the object of study.

Exploring TINL is of great importance for node representation learning due to its ubiquity and perniciousness. However, the methods [17, 22] for quantity imbalance can be hardly applied to TINL because of the orthogonality. To remedy the topology-imbalance issue, thus promoting the node classification, we propose a model-agnostic training framework **ReNode** to re-weight the labeled nodes according to their positions. We devise the conflic**t** detecti**o**n-based **To**pology **R**elative L**o**cation (**Totoro**) metric to leverage the interaction among labeled nodes across the whole graph to locate their structural positions. Based on the Totoro metric, we further increase the training weights of nodes with small conflict that are highly likely to be close to topological class centers to make them play a more pivotal role during training, and vice versa (right of Figure 1). Empirical results of various imbalance scenarios (TINL, QINL, large-scale graph) and multiple graph neural networks (GNNs) demonstrate the effectiveness and generalizability of our method. Besides, we provide the sensitivity to topology imbalance as a new evaluation perspective for different GNN architectures.

## 2 Topology-Imbalance Node Representation Learning

### 2.1 Notations and Preliminary

In this work, we follow the well-established semi-supervised node classification setting [47, 18] to conduct analyses and experiments. Given an undirected and unweighted graph $\mathcal{G} = (\mathcal{V}, \mathcal{E}, \mathcal{L})$, where $\mathcal{V}$ is the node set represented by the feature matrix $\boldsymbol{X} \in \mathbb{R}^{n*d}$ ($n = |\mathcal{V}|$ is the node size and $d$ is the node embedding dimension), $\mathcal{E}$ is the edge set which is represented by an adjacency matrix $\boldsymbol{A} \in \mathbb{R}^{n*n}$, $\mathcal{L} \subset \mathcal{V}$ is the labeled node set and usually we have $|\mathcal{L}| \ll |\mathcal{V}|$, the node classification

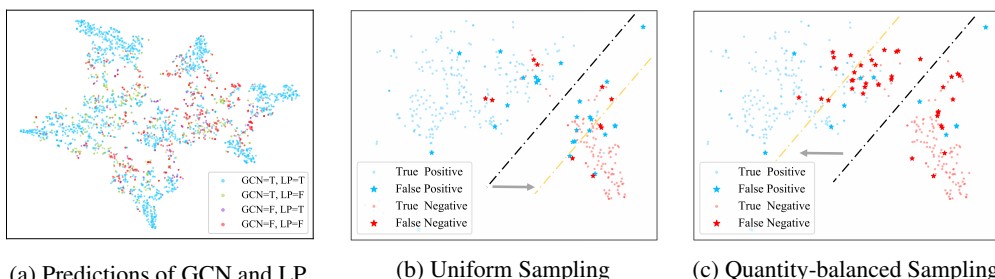

| (a) Predictions of GCN and LP | (b) Uniform Sampling | (c) Quantity-balanced Sampling |

Figure 2: Node influence and boundary shift caused by quantity- and topology-imbalance. (a): The prediction results of GCN and LP are highly consistent (t-SNE [39] visualization of the *CORA* dataset). (b): The node influence boundary (the yellow dotted line) is shifted towards the small class from the true class boundary (the black dotted line) under the quantity- and topology-imbalance scene. (c): The node influence boundary is shifted towards the large class under the quantity-balanced, topology-imbalanced scene. We regard the large class as positive class to indicate the results.

task is to train a classifier $\mathcal{F}$ (usually a GNN) to predict the class label $\mathbf{y}$ for the unlabeled node set $\mathcal{U} = \mathcal{V} - \mathcal{L}$. The training sets for different classes are represented by $(\mathcal{C}_1, \mathcal{C}_2, \cdots, \mathcal{C}_k)$ and $k$ is the number of classes. The labeling ratio $\delta = \mathcal{L}/\mathcal{V}$ is the proportion of labeled nodes in all nodes. In this work, we focus on TINL in homogeneously-connected graphs and hope to inspire future studies on the critical topology-imbalance issue.

## 2.2 Understanding Topology Imbalance via Label Propagation

From Figure 1, we can intuitively perceive the imbalance brought by the positions of labeled nodes; in this part, we further explore the nature of topology imbalance with the well-known Label Propagation [50] algorithm (short as LP) and provide a uniform analysis framework for the comprehensive node imbalance issue. In LP, labels are propagated from the labeled nodes and aggregated along edges, which can also be viewed as a random walk process from labeled nodes. The convergence result $\boldsymbol{Y}$ after repeated propagation is regarded as the nodes soft-labels:

$$\boldsymbol{Y} = \alpha(\boldsymbol{I} - (1-\alpha)\boldsymbol{A}')^{-1}\boldsymbol{Y}^0, \tag{1}$$

where $\boldsymbol{I}$ is the identity matrix, $\alpha \in (0, 1]$ is the random walk restart probability, $\boldsymbol{A}' = \boldsymbol{D}^{-\frac{1}{2}}\boldsymbol{A}\boldsymbol{D}^{-\frac{1}{2}}$ is the adjacency matrix normalized by the diagonal degree matrix $\boldsymbol{D}$, $\boldsymbol{Y}^0$ is the initial label distribution where labeled nodes are represented by the one-hot vectors. The prediction label for the $i$-th node is $\boldsymbol{q}_i = \arg\max_j \boldsymbol{Y}_{ij}$. LP is a simple yet successful model [37] and can be unified with GNN models owning the message-passing mechanism [41]. From Figure 2(a), we can empirically find that there is a significant correlation between the results of LP and GCN (T/F indicates prediction is True/False).

The LP prediction $\boldsymbol{q}$ can be viewed as the distribution of the *(labeled) node influence* [41] (i.e. each node is mostly influenced by which class's information); hence the boundaries of the node influence can act as an effective reflection for the GNN model decision boundaries considering the high consistency between LP and GNN. Moreover, node influence offers a unified view of TINL and QINL: ideally, the node influence boundaries should be consistent with the true class boundaries, but both the labeled nodes' numbers (QINL) and positions (TINL) can cause a shift of the node influence boundaries from the true one, resulting in deviation of the model decision boundaries.

**Node imbalance issue is composed of topology- and quantity-imbalance.** Figure 2 illustrates two examples of node influence boundary shift. In Figure 2(b), when the uniform selection is adopted to generate training set, both the quantity and the topology are imbalanced for model training; then the large class with more total nodes (denotes by blue color) will own stronger influence than the small class with fewer total nodes (denotes by red color) due to the quantity advantage and the node influence boundary is shifted towards the small class. In Figure 2(c), when the quantity-balanced strategy is adopted for sampling training nodes, it will be easier for the small class to has more labeled nodes close to the class boundary and the boundary of the node influence is shifted into the large class. We can find that even when the training set is quantity-balanced, the topology-imbalance issue still exists and hinders the node classification learning. Hence, we can conclude that node imbalance

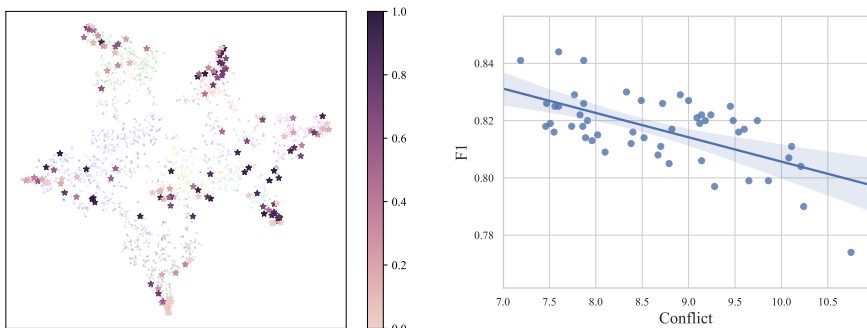

Figure 3: Effectiveness of Totoro at (a) Node Level: labeled nodes (t-SNE visualization of the *CORA* dataset) with less influence conflict (lighter color) are farther-away from class boundaries than those with high conflict (darker color), and (b) Dataset Level: There is a significant negative correlation between the GNN (GCN) performance and overall conflict of the training set (the Pearson correlation coefficient is $-0.618$ over 50 randomly selected training sets with the $p$ value smaller than 0.01).

learning is caused by the joint effect of TINL and QINL. Separately considering TINL or QINL will lead to a one-sided solution to node imbalance learning.

## 2.3 Measuring Topology Imbalance by Influence Conflict

Although we have realized that the imbalance of node topology interferes with model learning, how to measure the labeled node's relative topological position to its class (being far away from or close to the class center) remains the key challenge in handling the topology-imbalance issue due to the complex graph connections and the unknown class labels for most nodes in the graph. As the nodes are homogeneously connected when constructing the graph, even nodes close to the class boundaries own similar characteristics to their neighbors. Thus it is unreliable to leverage the difference between the characteristics of one labeled node and its surrounding subgraphs to locate its topological position. Instead, we propose to utilize the node topology information by considering the node influence conflict across the whole graph and devise the Conflic**t** Detecti**o**n-based **To**pology **R**elative L**o**cation metric (**Totoro**).

Similar to Eq (1), we calculate the Personalized PageRank [27] matrix $\boldsymbol{P}$ to measure node influence distribution from each labeled node:

$$\boldsymbol{P} = \alpha(\boldsymbol{I} - (1 - \alpha)\boldsymbol{A'})^{-1}. \tag{2}$$

**Node influence conflict denotes topological position.** According to related studies [41, 19, 2], $\boldsymbol{P}$ can be viewed as the distribution of influence exerted outward from each node. We assume that if a labeled node $v \in \boldsymbol{\mathcal{V}}$ encounters strong heterogeneous influence from the other classes' labeled nodes in the subgraph around node $v$ where node $v$ itself owns great influence, we have the conclusion that node $v$ meets large *influence conflict* in message passing and it is close to topological class boundaries, and vice versa. Based on this hypothesis, we take the expectation of the influence conflict between the node $v$ and the labeled nodes from other classes when node $v$ randomly walks across the entire graph as a measurement of how topologically close node $v$ is to the center of the class it belongs to. The Totoro value of node $v$ is computed as:

$$\boldsymbol{T}_v = \mathbb{E}_{x \sim \boldsymbol{P}_{v,:}}\big[ \sum_{j \in [1,k], j \neq \boldsymbol{y}_v} \frac{1}{|\boldsymbol{\mathcal{C}}_j|} \sum_{i \in \boldsymbol{\mathcal{C}}_j} \boldsymbol{P}_{i,x}\big], \tag{3}$$

where $\boldsymbol{y}_v$ is the ground-truth label of node $v$, $\boldsymbol{P}_v$ indicates the personalized PageRank probability vector for the node $v$. A larger Totoro value $\boldsymbol{T}_v$ indicates that node $v$ is topologically closer to class boundaries, and vice versa. The normalization item $1/|\boldsymbol{\mathcal{C}}_j|$ is added to make the influence from the different classes comparable when computing conflict.

We visualize the node labels and the Totoro values (scaled to $[0, 1]$) of labeled nodes in Figure 3(a). We can find that the labeled nodes with smaller Totoro values are farther away from the class boundaries, demonstrating the effectiveness of Totoro in locating the positions of labeled nodes. Besides, we sum

the conflict of all the labeled nodes $\sum_{b \in \mathcal{L}} \boldsymbol{T}_v$ to measure the overall conflict of the dataset, which can be viewed as the metric for the overall topology imbalance given the graph $\mathcal{G}$ and the training set $\mathcal{L}$. Figure 3(b) shows that there is a significant negative correlation between the overall conflict and the model performance, which further demonstrates the effectiveness of Totoro in measuring the intensity of topology imbalance at the dataset level.

## 2.4 Alleviate Topology Imbalance by Instance-wise Node Re-weighting

In this section, we introduce **ReNode**, a model-agnostic training weight schedule mechanism to address TINL for general GNN encoder in a plug-and-play manner. Inspired by the analysis in Section 2.2, the ReNode method is devised to promote the training weights of the labeled nodes that are close to the topological class centers, so as to make these nodes play a more active role in model learning, and vice versa. Specifically, we devise a cosine annealing mechanise [2] for the training node weights based on their Totoro values:

$$\boldsymbol{w}_v = w_{\min} + \frac{1}{2}(w_{\max} - w_{\min})(1 + \cos(\frac{\text{Rank}(\boldsymbol{T}_v)}{|\mathcal{L}|}\pi)), \quad v \in \mathcal{L} \tag{4}$$

where $\boldsymbol{w}_v$ is the modified training weight for the labeled node $v$, $w_{\min}, w_{\max}$ are the hyper-parameters indicating the lower bound and upper bound of the weight correction factor, $\text{Rank}(\boldsymbol{T}_v)$ is the ranking order of $\boldsymbol{T}_v$ from the smallest to the largest. The training loss $L_T$ for the quantity-balanced, topology-imbalanced node classification task is computed by the following equations:

$$L_T = -\frac{1}{|\mathcal{L}|} \sum_{v \in \mathcal{L}} \boldsymbol{w}_v \sum_{c=1}^{k} \boldsymbol{y}_v^{*c} \log \boldsymbol{g}_v^c, \quad \boldsymbol{g} = \text{softmax}(\mathcal{F}(\boldsymbol{X}, \boldsymbol{A}, \boldsymbol{\theta})), \tag{5}$$

where $\mathcal{F}$ denotes any GNN encoder, $\boldsymbol{\theta}$ is the parameter of $\mathcal{F}$, $\boldsymbol{g}_i$ is the GNN output for node $i$, $\boldsymbol{y}_i^*$ is the gold label for node $i$ in one-hot embedding. By encouraging the positive effects of the labeled nodes near the class topological centers, and reducing the negative effects of those near the topological class boundaries, our ReNode method is expected to minimize the deviation between the node influence boundaries and the true class boundaries, so as to correct the class imbalance caused by the positions of labeled nodes.

**ReNode to Jointly Handle TINL and QINL**   In this part, we introduce the application of the ReNode method in a more general graph imbalance scenario where both the topology- and quantity-imbalance issues exist. As analyzed in previous sections, the TINL and QINL are orthogonal problems. Therefore, we propose that our ReNode method based on (labeled) node topology can be seamlessly combined with the existing methods designed for the quantity-imbalance learning. Without loss of generality, we present how our ReNode method can be combined with the vanilla class frequency-based re-weight method [17]. The training loss $L_Q$ for the quantity-imbalanced, topology-imbalanced node classification task is formalized in the following equation:

$$L_Q = -\frac{1}{|\mathcal{L}|} \sum_{v \in \mathcal{L}} \boldsymbol{w}_v \frac{|\bar{\mathcal{C}}|}{|\mathcal{C}_j|} \sum_{c=1}^{k} \boldsymbol{y}_v^{*c} \log \boldsymbol{g}_v^c, \tag{6}$$

where $|\bar{\mathcal{C}}|$ is the average number of the class training sizes. With this method, the final weight of the labeled node is affected by two perspectives: training examples of the minority classes will have higher weights than that of the majority classes; training examples close to the topological class centers will have higher weights than those are close to the topological class boundaries.

**ReNode for Large-scale Graph**   There are mainly two challenges when applying ReNode to large-scale graphs: (1) how to calculate the PageRank matrix, and (2) how to train the GNN model in an inductive setting [13]. In this work, we follow the PPRGo method [2] to implement our method on the large-scale graph, which can decouple the feature learning process from the information transmission process to resolve the dependence on the global graph topology structure and can be carried out much efficiently. Following PPRGo, the Personalized PageRank matrix $\hat{\boldsymbol{P}}$ and the corresponding training ReNode factor $\hat{\boldsymbol{w}}$ are generated by the estimation method from Andersen et al. [1] and then $\hat{\boldsymbol{P}}$ is

---

[2]We also try other schedules and the cosine method works best. More details can be found in Appendix B.

Table 1: ReNode (short as RN) for the pure topology-imbalance issue. We report Weighted-F1 (W-F, %), Macro-F1 (M-F, %) and the corresponding standard deviation for each group of experiments. $*$ and $**$ represent the result is significant in student t-test with $p < 0.05$ and $p < 0.01$, respectively.

| Model | Training | CORA | | CiteSeer | | PubMed | | Photo | | Computers | |
|---|---|---|---|---|---|---|---|---|---|---|---|
| | | W-F | M-F | W-F | M-F | W-F | M-F | W-F | M-F | W-F | M-F |
| GCN | w/o RN | $79.1_{\pm1.1}$ | $77.8_{\pm1.5}$ | $66.2_{\pm1.0}$ | $62.0_{\pm1.3}$ | $74.6_{\pm2.1}$ | $74.7_{\pm1.9}$ | $86.8_{\pm2.0}$ | $84.7_{\pm1.7}$ | $74.2_{\pm2.6}$ | $73.6_{\pm2.9}$ |
| | w/ RN | $\mathbf{79.8}^{**}_{\pm0.9}$ | $\mathbf{78.6}^{**}_{\pm1.2}$ | $\mathbf{66.9}^{*}_{\pm1.1}$ | $\mathbf{62.8}^{*}_{\pm1.4}$ | $\mathbf{76.1}^{**}_{\pm1.5}$ | $\mathbf{76.1}^{**}_{\pm1.8}$ | $\mathbf{87.7}^{**}_{\pm2.2}$ | $\mathbf{85.4}^{**}_{\pm1.9}$ | $\mathbf{74.7}^{*}_{\pm2.2}$ | $\mathbf{74.5}^{*}_{\pm2.3}$ |
| GAT | w/o RN | $76.0_{\pm1.7}$ | $74.9_{\pm1.9}$ | $66.3_{\pm2.8}$ | $62.4_{\pm2.6}$ | $73.9_{\pm2.2}$ | $73.9_{\pm2.1}$ | $88.3_{\pm2.0}$ | $86.2_{\pm2.2}$ | $\mathbf{79.0}_{\pm2.1}$ | $\mathbf{78.8}_{\pm2.3}$ |
| | w/ RN | $\mathbf{77.7}^{**}_{\pm2.0}$ | $\mathbf{76.2}^{**}_{\pm1.8}$ | $\mathbf{67.1}^{*}_{\pm1.9}$ | $\mathbf{63.2}^{*}_{\pm1.6}$ | $\mathbf{75.2}^{**}_{\pm2.0}$ | $\mathbf{75.1}^{**}_{\pm2.5}$ | $\mathbf{89.1}^{**}_{\pm2.0}$ | $\mathbf{87.1}^{**}_{\pm2.0}$ | $78.8_{\pm1.9}$ | $78.7_{\pm2.0}$ |
| PPNP | w/o RN | $80.5_{\pm1.6}$ | $79.1_{\pm1.4}$ | $67.5_{\pm1.8}$ | $63.2_{\pm1.3}$ | $74.6_{\pm1.9}$ | $74.7_{\pm1.7}$ | $89.3_{\pm1.3}$ | $86.8_{\pm1.4}$ | $78.7_{\pm1.5}$ | $77.7_{\pm1.7}$ |
| | w/ RN | $\mathbf{81.9}^{**}_{\pm0.6}$ | $\mathbf{80.5}^{**}_{\pm0.8}$ | $\mathbf{68.1}^{*}_{\pm1.4}$ | $\mathbf{63.7}^{*}_{\pm2.0}$ | $\mathbf{76.0}^{**}_{\pm2.0}$ | $\mathbf{76.1}^{**}_{\pm2.2}$ | $\mathbf{89.7}^{*}_{\pm1.0}$ | $\mathbf{87.2}^{*}_{\pm1.3}$ | $\mathbf{79.0}^{*}_{\pm1.1}$ | $\mathbf{78.3}^{*}_{\pm1.1}$ |
| SAGE | w/o RN | $75.1_{\pm1.7}$ | $74.6_{\pm1.4}$ | $67.0_{\pm1.4}$ | $63.0_{\pm1.4}$ | $74.2_{\pm2.2}$ | $74.2_{\pm2.1}$ | $86.2_{\pm2.6}$ | $83.9_{\pm2.4}$ | $73.5_{\pm3.4}$ | $71.6_{\pm2.5}$ |
| | w/ RN | $\mathbf{75.7}^{**}_{\pm1.7}$ | $\mathbf{75.1}^{**}_{\pm1.4}$ | $\mathbf{67.3}_{\pm1.4}$ | $\mathbf{63.5}^{*}_{\pm1.2}$ | $\mathbf{74.9}^{**}_{\pm1.9}$ | $\mathbf{78.2}^{**}_{\pm2.3}$ | $\mathbf{86.5}_{\pm1.7}$ | $\mathbf{84.1}_{\pm1.7}$ | $\mathbf{74.9}^{**}_{\pm3.0}$ | $\mathbf{72.3}^{**}_{\pm2.5}$ |
| CHEB | w/o RN | $74.5_{\pm1.1}$ | $73.4_{\pm1.1}$ | $66.8_{\pm1.8}$ | $63.2_{\pm1.6}$ | $75.1_{\pm1.8}$ | $75.2_{\pm1.1}$ | $82.1_{\pm2.2}$ | $79.4_{\pm3.5}$ | $70.3_{\pm4.0}$ | $68.4_{\pm3.4}$ |
| | w/ RN | $\mathbf{75.3}^{**}_{\pm1.1}$ | $\mathbf{74.0}^{**}_{\pm1.1}$ | $\mathbf{67.5}^{**}_{\pm1.6}$ | $\mathbf{63.8}^{**}_{\pm1.5}$ | $\mathbf{76.2}^{**}_{\pm1.4}$ | $\mathbf{76.3}^{**}_{\pm1.2}$ | $\mathbf{84.8}^{**}_{\pm2.4}$ | $\mathbf{82.1}^{**}_{\pm2.8}$ | $\mathbf{70.5}_{\pm4.0}$ | $\mathbf{68.6}_{\pm3.4}$ |
| SGC | w/o RN | $74.9_{\pm2.1}$ | $73.8_{\pm2.1}$ | $65.7_{\pm1.6}$ | $61.8_{\pm1.6}$ | $72.9_{\pm2.3}$ | $73.1_{\pm2.6}$ | $87.1_{\pm1.3}$ | $84.9_{\pm1.1}$ | $77.4_{\pm1.7}$ | $76.8_{\pm1.8}$ |
| | w/ RN | $\mathbf{77.0}^{**}_{\pm1.1}$ | $\mathbf{76.0}^{**}_{\pm1.1}$ | $\mathbf{67.2}^{**}_{\pm1.3}$ | $\mathbf{62.9}^{**}_{\pm1.8}$ | $\mathbf{73.7}^{**}_{\pm2.8}$ | $\mathbf{73.8}^{**}_{\pm2.1}$ | $\mathbf{87.4}_{\pm1.5}$ | $\mathbf{85.2}_{\pm1.5}$ | $\mathbf{78.2}^{**}_{\pm1.8}$ | $\mathbf{77.8}^{*}_{\pm1.2}$ |

Table 2: Result of different dataset conflict levels (High/Middle/Low). Our ReNode method improve the GNN (GCN) performance most when the conflict level of graph is high.

| W-F(%) | CORA-H | CORA-M | CORA-L | CiteSeer-H | CiteSeer-M | CiteSeer-L | PubMed-H | PubMed-M | PubMed-L |
|---|---|---|---|---|---|---|---|---|---|
| w/o RN | $76.5_{\pm1.3}$ | $78.4_{\pm0.7}$ | $79.7_{\pm0.8}$ | $62.6_{\pm1.5}$ | $65.3_{\pm0.6}$ | $67.3_{\pm1.1}$ | $72.1_{\pm2.4}$ | $74.7_{\pm1.8}$ | $78.3_{\pm1.8}$ |
| w/ RN | $\mathbf{78.7}^{**}_{\pm0.8}$ | $\mathbf{79.3}^{**}_{\pm0.6}$ | $\mathbf{80.4}^{**}_{\pm0.6}$ | $\mathbf{63.8}^{**}_{\pm1.3}$ | $\mathbf{66.0}^{**}_{\pm0.8}$ | $\mathbf{67.5}_{\pm1.4}$ | $\mathbf{74.3}^{**}_{\pm2.1}$ | $\mathbf{75.6}^{**}_{\pm1.9}$ | $\mathbf{78.8}^{*}_{\pm1.5}$ |

directly employed as the aggregation weights from all the other nodes regardless of their topology distance from the current node:

$$g' = \mathrm{softmax}(\hat{\boldsymbol{P}}\mathcal{F}'(\boldsymbol{X},\theta')), \tag{7}$$

where $\mathcal{F}'$ can be a linear layer or a multi-layer perceptron with parameter $\theta'$. The final training loss for large-scale graph $L_L$ follows Eq (5) and (6), and replaces $\boldsymbol{w}$ and $\boldsymbol{g}$ with $\hat{\boldsymbol{w}}$ and $\boldsymbol{g}'$.

## 3 Experiments

In this section, we will first introduce the experimental datasets for both transductive and inductive semi-supervised node classification. Then we introduce the experiments to verify the effectiveness of the proposed ReNode method in three different imbalance situations: (1) TINL only, (2) TINL and QINL, (3) Large-scale Graph.

### 3.1 Datasets

We adopt two sets of graph datasets to conduct experiments. For the transductive setting [13], we take the widely-used Plantoid paper citation graphs [33] (*CORA*,*CiteSeer*, *Pubmed*) and the Amazon co-purchase graphs [24] (*Photo*,*Computers*) to verify the effectiveness of our method. For the inductive setting, we conduct experiments on the popular *Reddit* dataset [13] and the enormous *MAG-Scholar* dataset (coarse-grain version) [2] which owns millions of nodes and features. For each of these datasets, we repeat experiments on 5 different datasets splittings [34] and we run 3 times for each splitting to reduce the random variance. More details about the datasets and experiment settings are presented in Appendix A.

### 3.2 ReNode for the Pure Topology-imbalance Issue

**Settings** When considering topology-imbalance only, the labeling set takes a balanced setting and the annotation size for each class is all equal to $|\mathcal{L}|/k$. Following the most widely-used semi-supervised setting in node classification studies [47, 18], we randomly select 20 nodes in each class for training and 30 nodes per class for validation; all the remaining nodes form the test set. We display the experiment results for the 5 transductive datasets on 6 widely-used GNN models: GCN [18], GAT [40], PPNP [19], GraphSAGE [13] (short as SAGE), ChebGCN [9] (short as CHEB) and SGC [43]. We strictly align the hyperparameters in each group of experiments to show the pure improvement brought by our ReNode method (similarly hereinafter). The training loss $L_T$ from section 2.4 is adopted.

Table 3: ReNode method for the compound scene of TINL and QINL. The imbalance ratio $\rho$ is set to different levels ($[5, 10]$) to test the effect of our method under different imbalance intensities.

| Macro-F1(%) | CORA | | CiteSeer | | PubMed | | Photo | | Computers | |
|---|---|---|---|---|---|---|---|---|---|---|
| Imbalance Ratio | 5 | 10 | 5 | 10 | 5 | 10 | 5 | 10 | 5 | 10 |
| CE | $60.9_{\pm1.5}$ | $41.0_{\pm3.5}$ | $53.6_{\pm2.1}$ | $47.6_{\pm2.8}$ | $61.0_{\pm1.9}$ | $49.7_{\pm2.6}$ | $62.0_{\pm2.7}$ | $40.7_{\pm3.4}$ | $50.4_{\pm2.6}$ | $35.5_{\pm3.2}$ |
| DR-GCN | $67.7_{\pm1.1}$ | $51.3_{\pm1.4}$ | $54.7_{\pm1.7}$ | $52.5_{\pm2.6}$ | $79.4_{\pm1.2}$ | $78.0_{\pm1.6}$ | $80.8_{\pm2.3}$ | $79.5_{\pm2.8}$ | $66.9_{\pm3.5}$ | $67.4_{\pm3.6}$ |
| RA-GCN | $69.0_{\pm1.5}$ | $51.7_{\pm1.7}$ | $\mathbf{55.6}_{\pm1.3}$ | $52.7_{\pm2.1}$ | $80.6_{\pm1.8}$ | $78.1_{\pm2.1}$ | $81.4_{\pm2.6}$ | $79.4_{\pm3.2}$ | $71.2_{\pm2.8}$ | $68.7_{\pm3.0}$ |
| G-SMOTE | $68.1_{\pm0.9}$ | $49.6_{\pm1.1}$ | $54.0_{\pm1.6}$ | $51.8_{\pm1.3}$ | $79.7_{\pm1.2}$ | $76.4_{\pm1.5}$ | $82.2_{\pm1.8}$ | $77.5_{\pm2.1}$ | $71.9_{\pm2.5}$ | $61.3_{\pm3.2}$ |
| RW (w/o RN) | $69.1_{\pm1.4}$ | $49.7_{\pm1.6}$ | $53.6_{\pm2.3}$ | $52.9_{\pm2.6}$ | $80.5_{\pm1.5}$ | $78.0_{\pm2.0}$ | $80.5_{\pm2.7}$ | $80.4_{\pm3.3}$ | $70.5_{\pm3.2}$ | $67.8_{\pm4.2}$ |
| RW (w/ RN) | $70.0^{*}_{\pm1.3}$ | $50.1_{\pm1.7}$ | $55.2^{**}_{\pm1.8}$ | $54.0^{**}_{\pm2.5}$ | $81.2^{*}_{\pm1.0}$ | $78.5^{*}_{\pm2.2}$ | $83.9^{**}_{\pm2.1}$ | $\mathbf{81.3}^{**}_{\pm3.2}$ | $72.4^{**}_{\pm2.6}$ | $\mathbf{70.2}^{**}_{\pm2.4}$ |
| FOCAL (w/o RN) | $66.4_{\pm1.6}$ | $51.9_{\pm1.8}$ | $54.3_{\pm1.3}$ | $54.0_{\pm1.9}$ | $80.5_{\pm0.7}$ | $78.0_{\pm1.6}$ | $79.3_{\pm1.9}$ | $79.2_{\pm2.2}$ | $65.8_{\pm2.7}$ | $63.9_{\pm2.6}$ |
| FOCAL (w/ RN) | $68.7^{**}_{\pm0.7}$ | $52.6^{**}_{\pm1.9}$ | $54.6_{\pm1.2}$ | $\mathbf{54.7}^{*}_{\pm1.5}$ | $80.9^{*}_{\pm0.8}$ | $\mathbf{78.7}^{**}_{\pm1.4}$ | $80.0^{**}_{\pm2.3}$ | $80.7^{**}_{\pm2.9}$ | $68.6^{**}_{\pm3.1}$ | $65.5^{**}_{\pm3.5}$ |
| CB (w/o RN) | $69.8_{\pm1.5}$ | $51.5_{\pm1.5}$ | $54.1_{\pm1.3}$ | $53.5_{\pm0.8}$ | $80.6_{\pm0.8}$ | $77.6_{\pm1.6}$ | $77.9_{\pm2.6}$ | $78.8_{\pm3.1}$ | $69.6_{\pm2.2}$ | $64.8_{\pm2.9}$ |
| CB (w/ RN) | $\mathbf{71.1}^{**}_{\pm0.6}$ | $51.9^{*}_{\pm1.2}$ | $54.7^{*}_{\pm1.6}$ | $54.3^{**}_{\pm2.3}$ | $81.2^{*}_{\pm1.8}$ | $78.3^{**}_{\pm2.6}$ | $79.6^{**}_{\pm2.7}$ | $80.4^{**}_{\pm3.3}$ | $\mathbf{73.1}^{**}_{\pm3.1}$ | $66.5^{**}_{\pm3.6}$ |

**Results** From Table 1, we can find that our ReNode method can effectively improve the overall performance (Weighted-F1) and the class-balance performance (Macro-F1) for all the 6 experiment GNNs in most cases, which proves the effectiveness and generalizability of our method. Our method considers the graph-specific topology imbalance issue which has been usually neglected in existing methods and conducts a fine-grained and self-adaptive adjustment to the training node weights based on their topological positions. We notice that the improvement for the *CiteSeer* dataset is less than the other datasets. We analyze the reason lies in that the connectivity of *CiteSeer* is poor, which makes the conflict detection–based method fail to reflect the node topological position well. To verify the motivation of relieving topology-imbalance, we set training sets with different levels of topology-imbalance to test our method[3]. Table 2 displays that our ReNode method improves the performance of GNN (GCN) most when the dataset is highly topologically imbalanced, which demonstrates that our method can effectively alleviate topology-imbalance and improve GNN performance.

## 3.3   ReNode for the Compound Scene of TINL and QINL

**Settings** When jointly considering both topology- and quantity-imbalance issues, following existing studies [5, 4], we take the step imbalance setting, in which all the minority classes have the same labeling size $n_i$ and all the majority classes have the same labeling size $n_a = \rho * n_i$. The imbalance ratio $\rho$ denotes the intensity of quantity imbalance which is equal to the ratio of the node size of the most frequent to least frequent class. In this work, the imbalance ratio $\rho$ is set to $[5, 10]$ for each dataset. The fraction of the majority classes is $\mu$, and for all experiments, we set $\mu = 0.5$ and round down the result $\mu * k$. The training loss $L_Q$ from section 2.4 is adopted. We implement two groups of baselines for comparison: (1) Popular quantity-imbalance methods for general scenarios: Re-weight [17] (RW), Focal Loss [22] (Focal) and Class Balanced Loss [8] (CB); (2) Graph-specific quantity-imbalance methods: DR-GCN [35], RA-GCN [11] and GraphSMOTE [49]. To jointly handle the topology- and quantity-imbalance issues and demonstrate the orthogonality of them, we combine our ReNode method with these three general quantity-imbalance methods (RW, Focal, CB)[4]. The backbone model is GCN [18], and the labeling ratio $\delta$ is set to $5\%$.

**Results** From Table 3 (Macro-F1 is reported here for a fair comparison with these methods designed for class-balance performance), we can find that our ReNode method significantly outperforms both the general and the graph-specific quantity-imbalance methods in most situations by simultaneously alleviating the topology- and quantity-imbalance issues. Even when the training set is severely quantity-imbalanced ($\rho$=10), our method still effectively alleviates the imbalance issue and promotes model performance well. The performance of the quantity-imbalance methods from the general field (RW, Focal, CB) is on par with or less effective than the graph-specific quantity-imbalance methods (DR-GCN, RA-GCN, G-SMOTE), while the combination of our ReNode method and these general quantity-imbalance methods can surpass the graph-specific quantity-imbalance methods, which demonstrates that the node imbalance learning can be further solved by jointly handling the topology- and quantity-imbalance issues instead of considering the quantity-imbalance issue only.

---

[3]The settings of the dataset topology-imbalance levels is shown in Appendix C.

[4]The three graph-specific methods have special operations for the training loss, which can be hardly combined with our method.

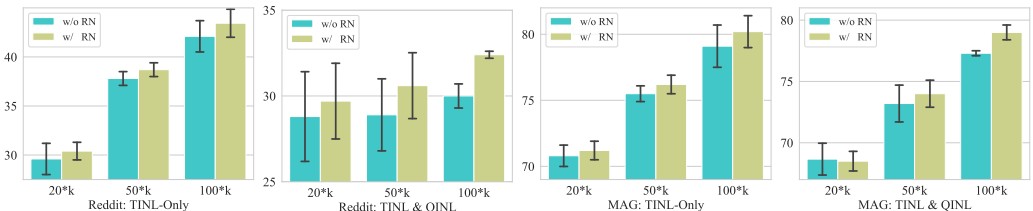

Figure 4: Experimental results (Weighted-F1,%) on the large-scale *Reddit* and *MAG-Scholar* graphs. Our ReNode method can effectively improve the model performance under different labeling sizes.

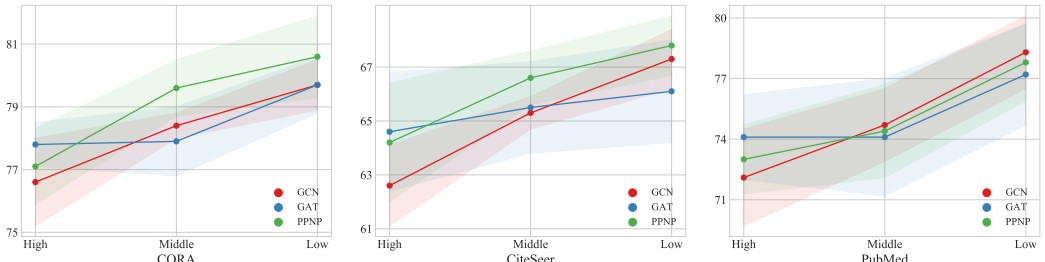

Figure 5: Evaluating GNNs from the aspect of topology-imbalance sensitivity (Metric: Weighted-F1 (%)). We can summarize the ranking of topology-imbalance sensitivity: GCN > PPNP > GAT.

## 3.4 ReNode for Large-scale Graphs

**Settings** We conduct experiments on the two large-scale datasets: Reddit and MAG-Scholar, to verify the effectiveness of our ReNode method in the inductive setting. We conduct experiments with different labeling sizes (20/50/100 training nodes per class) and imbalance settings (TINL-only, TINL and QINL). The backbone GNN model is PPRGo [2] [5]. For QINL, we take the uniform selection to sample training nodes to be consistent with PPRGo. The training loss $L_L$ from section 2.4 is adopted. Both baseline and our methods are not combined with any quantity-imbalance method.

**Results** In Figure 4, we present the experiment results with different labeling sizes and imbalance settings, we can find that our method can effectively promote the performance on the large-scale graphs comparing to the popular PPRGo model across different settings, which demonstrates the applicability of our method for extremely-large graphs. We also notice that our method can bring greater improvement when the labeling size is large. We explain the reason lies in that when the labeling size is large, the positions located by the conflicts among nodes will be more accurate, thus bringing more reasonable weight adjustments. On the other hand, when the labeling ratio is extremely small (especially for the enormous *MAG-Scholar* graph) and the influence conflict between the labeled nodes is negligible, our method exhibits the cold start problem.

## 4 Discussions

### 4.1 Evaluating GNNs from the Aspect of Topology-Imbalance Sensitivity

In Figure 5, we evaluate the GNN's capability for handling topology-imbalance and find that different GNNs present significant difference in the topology-imbalance sensitivity across multiple datasets. The GCN model is susceptible to the topology-imbalance level of the graph and its performance decays greatly when the topology-imbalance increases. On the opposite, the GAT model is less sensitive to the topology-imbalance level and can achieve the best results when the topology-imbalance level is high. The PPNP model can achieve ideal performance when the topology-imbalance level is low, and its performance does not drop as sharply as GCN when the topology-imbalance level is high. We analyze the reason lies in that: (1) the aggregation operation of GCN is equivalent to directly

---

[5]We do not implement other inductive backbone GNN models like Cluster-GCN [7] or APPNP [19] due to the low efficiency [2] of them when handling the enormous *MAG-Scholar* dataset.

averaging neighbor features [45] that lacks the noise filtering mechanism, so it is more sensitive to the topology-imbalance level of the graph; (2) the GAT model can dynamically adjust the aggregation weight from different neighbors, which increases its robustness to the high topology-imbalance situation but hinders the model performance when the graph topology-imbalance level is low and there is less need to filter neighbor information; (3) the infinite convolution mechanism of the PPNP model makes it possible to aggregate the information from distant nodes to enhance its robustness to the graph topology imbalance.

Shchur et al. [34] notice that the performance ranking of GNNs varies with the training set selection. Hence, existing node classification studies [32, 14] usually repeat experiments multiple times with different training sets to reduce this randomness. The results from Figure 5 inspire us that the topology imbalance can partly explain the randomness of GNN performance caused by the training set selection and we can adopt the topology-imbalance sensitivity as a new aspect in evaluating the performance of different GNN architectures.

### 4.2 Limitations of Method

Although our ReNode method has proven effective in multiple scenarios, we also notice some limitations of it because of the complexity of node imbalance learning. First, the ReNode method is devised for homogeneously-connected graphs (linked nodes are expected to be similar, such as the various datasets in experiments), and it needs a further update for heterogeneously-connected graphs (such as protein networks). Besides, the ReNode method improves less when the graph connectivity is poor (Section 3.2) or the labeling ratio is extremely low (Section 3.3) because in these cases, the conflict level among nodes is low thus the nodes topological positions are insufficiently reflected.

## 5 Related Work

Imbalanced classification problems are widespread in real scenarios and have attracted extensive attention from both academia and industry. Most existing studies on this topic focus on the class-imbalanced quantity distribution [15], where the model's inference ability for the majority classes will be significantly better than that of minority classes [12]. The existing methods for solving the quantity-imbalance issue can be roughly divided into methods for the data selection phase and the model training phase. Active learning [31, 10, 42] and Re-sampling [6, 16, 25] are two classical examples designed to construct a quantity-balanced training set . On the other hand, Re-weighting is a simple but effective solution for the model training phase, which adjusts the weights of training samples in different classes based on the labeling sizes [17, 30, 8, 5]. However, directly applying these methods into the graph scene lacks the consideration for the graph-specific topology-imbalance issue. Unlike the re-weight methods which conduct class-lever re-weighting, our ReNode method is a more fine-grained one and assign weights to each node individually.

There have been quantity-imbalance studies (Tomek links [38], NearMiss [23], One-Sided Selection [20]) trying to exclude the negative influence of labeling samples close to class boundaries by measuring the similarity of sample features. However, in the graph scene, the prior knowledge contained in node connections is more reliable than directly calculating the feature similarity. Besides, the number of labeled nodes is quite small in the semi-supervised setting. Thus it is not robust to locate their positions by computing similarity among a small number of nodes and we propose to leverage the influence conflict across the whole graph to locate node position to boundaries.

Graph data structure owns a wide range of applications, such as social media [13], stock exchange [21], shopping [34], medicine [44], transportation [28] and so on. Similar to other data structures, graph node representation learning also suffer from the quantity-imbalance issue [35]. Apart from the universal quantity-balance approaches introduced in Section 5 which can be transferred to the graph scene, there are some graph-specific quantity-imbalance methods recently proposed. DR-GCN [35] propose two types of regularization to tackle quantity imbalance: class-conditioned adversarial training and unlabeled nodes latent distribution constraint. RA-GCN [11] propose to automatically learn to weight the training samples in different classes in an adversarial training manner. AdaGCN Shi et al. [36] propose to leverage the boosting algorithm to handle the quantity-imbalance issue for the node classification task. GraphSMOTE [49] combines the synthetic node generation and the edge generation to up-sample nodes for the minority classes. However, these studies only pay attention to the quantity imbalance and overlook the topology imbalance.

Different from these studies [48, 29, 26] that try to locate the absolute positions for all the nodes by measuring their distance from the selected anchor nodes, our Totoro metric is devised to locate the relative positions to the class boundary for the labeling nodes by considering the influence conflict and can get rid of the dependence on the anchor nodes. Besides, our relative positions can more accurately reflect node class information because we distinguish the information from different classes while existing studies [48, 29] treat all the anchor nodes the same and ignore the class difference.

## 6 Conclusion and Future Work

In this work, we recognize the topology-imbalance node representation learning (TINL) as a graph-specific imbalance learning problem that has not been studied so far. We find that the topology-imbalance issue widely exists in graphs and severely hinders the learning of node classification. We unify TINL with the quantity-imbalance node representation learning (QINL) by considering the shift of the node influence boundaries from true class boundaries. To measure the degree of topology imbalance, we devise a conflict detection–based metric Totoro to locate node position, and further propose the ReNode method to adaptively adjust the training weights of labeled nodes based on their topological positions. Extensive empirical results have verified the effectiveness of our method in various settings: TINL-only, both TINL and QINL, and large-scale graph. Besides, we also propose the topology-imbalance sensitivity as a new metric to evaluate GNNs.

Considering the importance of the topology-imbalance issue and the limitations of our approach, advanced methods with stronger theoretical or experimental support are expected in future work. Moreover, since topology imbalance is widespread in graph-related tasks other than node classification, how to measure and solve the topology-imbalance issues in broader graph scopes remains a meaningful challenge for future study.

## 7 Acknowledgement

We appreciate all the thoughtful and insightful suggestions from reviews. This work was supported in part by a Tencent Research Grant and National Natural Science Foundation of China (No. 61673028). Xu Sun is the corresponding author of this paper.

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
