# Appendix for Topology-Imbalance Learning for Semi-Supervised Node Classification

**Deli Chen**[1,2], **Yankai Lin**[1], **Guangxiang Zhao**[2], **Xuancheng Ren**[2],
**Peng Li**[1], **Jie Zhou**[1], **Xu Sun**[2]
[1]Pattern Recognition Center, WeChat AI, Tencent Inc., China
[2]MOE Key Lab of Computational Linguistics, School of EECS, Peking University
{delichen, yankailin, patrickpli,withtomzhou}@tencent.com
{zhaoguangxiang,renxc,xusun}@pku.edu.cn

## A More Details of Dataset and Experiments

**Linux Server**    We run all the experiments on a Linux server, some important information is listed:

- CPU: Intel(R) Xeon(R) Silver 4210 CPU @ 2.20GHz $\times$ 40
- GPU: NVIDIA GeForce RTX2080TI-11GB $\times$ 8
- RAM: 125GB
- cuda: 11.1

**Python Package**    We implement all deep learning methods based on Python3.7. The experiment code is included in the supplementary materials. The versions of some important packages are listed:

- torch [11]: 1.9.1+cu111
- torch-geometric [4]: 2.0.1
- torch-cluster:1.5.9
- torch-sparse: 0.6.12
- scikit-learn: 1,0
- numpy:1.20.3
- scipy:1.7.1

**Datasets**    The statistical information of our experimental datasets are shown in Table 1. All these data are publicly available and the URLs listed as follows:

- Planetoid Citation Datasets [13] (*CORA/CiteSeer/PubMed*): https://github.com/rusty1s/pytorch_geometric/blob/master/torch_geometric/datasets/planetoid.py
- Amazon Co-purchasing Datasets [9] (*Photo/Computers*): https://github.com/rusty1s/pytorch_geometric/blob/master/torch_geometric/datasets/amazon.py
- Reddit Comment Dataset [5]:https://github.com/TUM-DAML/pprgo_pytorch/blob/master/data/get_reddit.md
- MAG-Scholar Dataset [1] (coarse grained version): https://figshare.com/articles/dataset/mag_scholar/12696653/2

35th Conference on Neural Information Processing Systems (NeurIPS 2021), Sydney, Australia.

Table 1: Statistical information about datasets. M indicates million.

| Dataset | #Node | #Edge | #Feature | #Class | Training |
|---------|-------|-------|----------|--------|----------|
| CORA | 2,708 | 5,429 | 1,433 | 7 | Transductive |
| CiteSeer | 3,327 | 4,732 | 3,703 | 6 | Transductive |
| PubMed | 19,717 | 44,338 | 5,00 | 3 | Transductive |
| Photo | 7,487 | 119,043 | 745 | 8 | Transductive |
| Computers | 13,381 | 245,778 | 767 | 10 | Transductive |
| Reddit | 232,965 | 11,606,919 | 602 | 41 | Inductive |
| MAG-Scholar | 10.5145M | 132.8176M | 2.7842M | 8 | Inductive |

Table 2: Dataset Topology-Imbalance Level

| $\sum_{v \in \mathcal{L}} T_v$ | LOW | MIDDLE | HIGH |
|---------|-----|--------|------|
| CORA | $4.26_{\pm 0.27}$ | $6.03_{\pm 0.21}$ | $7.39_{\pm 0.43}$ |
| CiteSeer | $1.19_{\pm 0.11}$ | $2.26_{\pm 0.01}$ | $4.37_{\pm 0.23}$ |
| Pubmed | $0.14_{\pm 0.02}$ | $0.25_{\pm 0.01}$ | $0.42_{\pm 0.05}$ |

**Dataset Splitting**    In training, we run 5 different random splittings for each dataset to relieve the randomness introduce by the training set selection following Shchur et al. [14]. We repeat experiments 3 times for each splitting to relieve the training splitting. The final performance (weighted F1, macro F1, and the standard deviation) is calculated based on the 15 repeated experiments. The dataset splitting seed list is $[0, 1, 2, 3, 4]$; the model training random seed list is: $[0, 1, 2]$.

**Method Hyperparameters**    For all encoders ($\mathcal{F}$ and $\mathcal{F}'$), we stacked two GNN or linear layers with the ReLU [10] activation function[1]. All the hyper-parameters are tuned on the validation set. The tuning range of dataset-specific hyperparameters is as follows:

- PageRank teleport probability $\alpha$: $[0.05, 0.1, 0.15, 0.2]$;
- Dimension of hidden layer: $[16, 32, 64, 128, 256]$;
- Lower bound of the cosine annealing $w_{min}$: $[0.25, 0.5, 0.75]$;
- Upper bound of the cosine annealing $w_{max}$: $[1.25, 1.5, 1.75]$;

**Training Setting**    We take the Adam [7] as the model optimizer. The learning ratio begins to decay after 20 epochs with a ratio of 0.95. We early stop the training process if there is no improvement in 20 epochs. The tuning range of dataset-specific hyperparameters is as follows:

- Learning Rate:$[0.005, 0.0075, 0.01, 0.015]$,
- Dropout Probability: $[0.2, 0.3, 0.4, 0.5, 0.6]$.

## B   Supplement to the ReNode Method

Apart from the relative ranking re-weight method in ReNode, we also tried to adjust the training weight based on the following scheduling methods:

- Linear decay based on the original node Totoro values;
- Linear decay based on the rank of node Totoro values;
- Discrete values for different nodes with a piece-wise function;

Among all these methods, the presented cosine annealing method works best. We analyze the reason lies in that, PageRank is proposed for node ranking; hence adjusting weights based on the original

---

[1]Except the SGC model, which increases the power iteration times of the normalized adjacency matrix to replace stacking GNN layers.

values is not robust and can be largely affected by outliers. Comparing to the linear decay schedule, the cosine schedule methods pay more attention to nodes with middle-level conflict, distinguishing which is of great importance for the model training.

The ReNode method assigns more weights to nodes far away from the graph class boundaries, which it is different from methods used in metric learning [15, 3] or contrastive learning [6, 12] that pay more attention to the 'hard' samples closing to class boundaries. In semi-supervised node classification, most message-passing based GNN model (e.g. GCN) relies on smoothing the adjacent nodes to transfer the category information from the labeled nodes to the unlabeled nodes [8, 2]. Thus, the 'easy' labeled nodes far away from the class boundaries are expected to better represent class prototypes. Enlarging the training weights of those 'hard' nodes that are close to the class boundaries makes it easier to confuse the class prototype with others. Besides, the labeling size in semi-supervised learning is much smaller than supervised learning (usually 20 nodes per class) and usually, the training nodes are sampled randomly. Hence, a very likely scenario is that the 'hard' samples for some categories are very close to the true class boundaries, while the 'hard' samples for other categories are far away from the true class boundaries. Relying on these 'hard' nodes to decide decision boundaries will cause a large shift of decision boundaries from the true ones.

## C   Settings of Dataset Topology-Imbalance Levels

In Section 3, we evaluate the model performance under different levels of topology imbalance. We introduce the settings for the topology-imbalance levels. For each experiment dataset, we randomly sampled 100 training sets, and calculate the dataset overall conflict as introduced in Section 2.3. Then we choose the 3 training sets with the highest/middle/lowest overall conflict as the high/middle/low-level topology-imbalance setting and report the average results on the 3 training sets for each dataset. The specific conflict values of different levels are displayed in Table 2.