# OpenReview forum: "Topology-Imbalance Learning for Semi-Supervised Node Classification"
_NeurIPS.cc/2021/Conference — NeurIPS 2021 Poster_

### Official Review · Reviewer_t36c · 2021-07-15

**Rating:** 6
**Confidence:** 4

**Summary:**

This paper first points out the topology-imbalance problem of node representation learning on graph-structured data. To address the problem (which is denoted as Topology-Imbalance Node Representation Learning, TINL), this paper presents a conflict detection-based metric (named Totoro) based on label propagation algorithm and Personalized PageRank matrix. Then, the authors introduce the ReNode algorithm that considers the Totoro of each node in semi-supervised training of GNN models. The ample experimental results show the superiority of ReNode in addressing the TINL problem.

**Limitations And Societal Impact:**

The authors have addressed the limitation and potential negative societal impact of the paper.

**Main Review:**

Originality: This paper introduces a novel research problem of semi-supervised node classification on graphs: the topology-imbalance problem. A minor concern is that the proposed solution mostly relies on the existing techniques (label propagation, PPR, and weighted training loss), although the combination of them is reasonable.

Quality: The paper has good quality. The proposed problem is critical and meaningful in real-world model training. The proposed Totoro metric is highly related to the research problem. The proposed ReNode algorithm is flexible and easy to apply. The experimental results can well evaluate the effectiveness of the proposed method.

Clarity: The paper is well-organized and has a good writing style. The motivation, proposed method, experiments, and analysis are all introduced clearly in the paper. There are some minor problems, e.g. in Equation (3), the definition of P_v is not given very clearly in the paper.

Significance: This paper addresses the topology-imbalance problem in semi-supervised node classification, which is different from most existing works addressing the quantity-imbalance problem. I believe that pointing out this new research problem is significant to the community. The proposed method can well handle the problem, which is proved by the empirical results.

Problems: I have two further concerns about this paper:
1.Although the performance of the proposed method is validated by experiments, the method lacks a theoretical explanation or boundary to prove its effectiveness. Concretely, the designs of Totoro metric and ReNode method seem experience-based but do not have enough theory to support them.
2.The computation of Totoro and node weights is based on the known labelled nodes, while these labels themselves may be biased due to the topology-imbalance problem. Hence, how to prove that the biased labels would generate unbiased node weights?

Post Rebuttal:
The responses answer my questions, and the authors have revised the paper according to my previous opinions. I do not have further questions. Based on the quality I will remain my rating.


**Time Spent Reviewing:**

4

---

> ### Author Response · Authors · 2021-08-08
> **Response to Official Review of Paper164 by Reviewer t36c**
>
> Response to Official Review of Paper164 by Reviewer t36c
> NeurIPS 2021 Conference Paper164 AuthorsDeli Chen(privately revealed to you)
> Official Comment by Paper164 Authors08 Aug 2021 (modified: 08 Aug 2021) Program Chairs, Paper164 Senior Area Chairs, Paper164 Area Chairs, Paper164 Reviewers Submitted, Paper164 Authors
> Comment:
>
> Thanks for your insightful comments on our work! Here is my response to the concerned questions.
>
> [Q1] Response to "the method lacks a theoretical explanation"
>
> Yes, the main intention of this work is to point out the problem of Topology Imbalance, a highly ubiquitous, pernicious but ignored issue in graph study. Both Totoro and ReNode are devised for this intention and the empirical results have proven their effectiveness under multiple graph datasets and GNN models, various imbalanced scenes (TINL-only, TINL&QINL, large-scale graphs), and strict repeat experiment settings (random dataset splitting with seeds [0,1,2,3,4] and random experiment initialization with seeds [0,1,2]). We also expect further improved methods can be proposed for the essential but ignored topology imbalance issue in the future, including both theoretical and experimental ones.
>
> \
> [Q2] Response to the "how to prove that the biased labels would generate unbiased node weights"
>
> Our method can be regarded as sorting all the labeled nodes according to their influence conflicts, thereby identifying nodes relatively close to the category centers and increasing their positive effects on model learning, and vice versa. In fact, we don't know which nodes are unbiased, but **we know which nodes are less biased to others**. Therefore, although random data partitioning will bring varying degrees of topology imbalance to the training set, our method can alleviate the topological imbalance issue consistently by assigning more weights to those relatively unbiased labeled nodes.
>
> \
> [Q3] Response to "the proposed solution mostly relies on the existing techniques, although the combination of them is reasonable"
>
> As the first study on the challenging topology imbalance issue, we want to provide a lite and general solution that can serve as a reliable reference for future studies. Designing methods based on the widely used LP and re-weighting makes our method has good compatibility with many popular GNNs (GCN, GAT, GraphSAGE, SGC, and so on) and brings limited additional computational cost.
> Besides, directly transferring existing techniques (LP, re-weighting) into the new scenario of topology imbalance can hardly work and the design of our method is none-trivial: unlike quantity imbalance which can be directly measured by metric like imbalance ratio (max class num/min class num), topology imbalance can be hardly measured directly thus designing the metric for it is quite challenging; we creatively propose to take the node influence conflict as an indirect reflection of topology imbalance, which has proved effective both intuitively (Figure 3) and empirically (Table 2).
>
> \
> [Q4] Response to "the definition of P_v is not given very clearly "
>
> Thank you for pointing out this. P_v represents the personalized PageRank probability vector for the node v. We will carefully check the paper and refine these writing problems.

---

### Official Review · Reviewer_1Cph · 2021-07-15

**Rating:** 5
**Confidence:** 5

**Summary:**

The paper studied the problem of imbalanced classification on graphs, by proposing to unify the quantity imbalance and topology imbalance together. The authors develop a label propagation algorithm that locates the topological position of the labeled nodes, and then re-weight nodes to alleviate the imbalanced data distribution. Experimental results on various benchmark datasets show the performance of the proposed model.

**Limitations And Societal Impact:**

Yes

**Main Review:**

The paper is easy-to-follow and well-organized. The approach and the solution to the problem are very well handled. Despite this, there lacks some insightful analysis about the proposed approach, and the empirical studies should be further improved. Here are the detailed comments:

[Related work] The idea of using label propagation for imbalance classification on graphs is not novel, such as [1]. The authors may want to re-conduct the literature review in the context of imbalanced classification and discuss the difference between this paper and the previous work.
[1] https://dl.acm.org/doi/10.1145/3219819.3219968

[Clarity] The authors stated that the imbalance classification problem “roots from the unequal quantity of labeled examples”, which is misleading and vulnerable. In general, in the setting of imbalanced classification, both the labeled set and the unlabeled set exhibit highly skewed data distribution.

[Technical contribution] The technical contribution of this paper is unclear – both the idea of label propagation and node-reweighting scheme are not novel in the setting of imbalance classification. The authors may want to provide more discussion to show the technical contribution of this paper.

[Evaluation] In Tab.1, the comparison experiments are conducted on CORA, Citeseer and PubMed, which are standard (balanced) node classification benchmark datasets. I don’t see any reasons why they are considered for imbalanced node classification. Moreover, all of the baselines (GCN, GAT, SGC, SAGE, CHEB) are not designed for imbalanced classification. With these, I doubt the comparison results in Tab.1 are unfair and do not stand.


**Time Spent Reviewing:**

1 hour

---

> ### Author Response · Authors · 2021-08-08
> **Response to Official Review of Paper164 by Reviewer 1Cph**
>
> Thanks for your detailed comments! Here is our response to the concerned questions.
>
> [Q1] Response to the novelty from existing imbalance classification studies.
>
> The main intention of this work is to point out the problem of Topology Imbalance, a highly ubiquitous, pernicious but ignored issue in graph study. A basic view of our work is that **quantity and topology imbalance are orthogonal problems** in graph scenes (Section 1). We conducted a detailed literature survey before the start of this work. So far, all the previous studies (including the mentioned one) related to graph imbalance learning focus on the topic of quantity imbalance, which is fundamentally different from our research which focuses on the graph-specific topology imbalance: topology imbalance considers the imbalance of the topological location distribution among different classes, while quantity imbalance considers the imbalance of the quantity distribution among different classes. Hence, the task objectives, dataset settings, and baseline models are different in these two imbalance scenarios.
>
> \
> [Q2] Response to " both the idea of label propagation and node-reweighting scheme are not novel in the setting of imbalance classification"
>
> As the first study on the challenging topology imbalance issue, we want to provide a lite and general solution that can serve as a reliable reference for future studies. Designing methods based on the widely used LP and re-weighting makes our method has good compatibility with many popular GNNs (GCN, GAT, GraphSAGE, SGC, and so on) and brings limited additional computational cost.
>
> Besides, according to our analysis, topology imbalance is orthogonal to quantity imbalance (Section 1). Even if the training set is quantity balanced, it still can be topology imbalanced (Section 2.1); therefore, the existing quantity-imbalance methods are not suitable for the topology-imbalance issue. Directly transferring existing techniques (LP, re-weighting) into the new scenario of topology imbalance can hardly work. The design of our method is none-trivial and achieves superior results in the topology imbalance scene: unlike quantity imbalance which can be directly measured by metric like imbalance ratio (max class num/min class num), topology imbalance can be hardly measured directly thus designing the metric for it is quite challenging; we creatively propose to take the PPR conflict as an indirect reflection of topology imbalance, which has proved effective both intuitively (Figure 3) and empirically (Table 2). Then we adopt the conflict degree to guide the training loss reweighting in an element-wise manner, which effectively releases the topology imbalance issue in various settings (TINL-only, TINL&QINL, large-scale graphs).
>
>
> \
> [Q3] Response to "both the labeled set and the unlabeled set exhibit highly skewed data distribution"
>
> It is true that the quantity distribution of both labeled and unlabeled sets is uneven. What we mean by this sentence is that in quantity-imbalance learning, the model's preference for the majority class comes from the unequal number of training set during the training process. In fact, the quantity-imbalance intensity is usually measured by the ratio of the node size of the most frequent to the least frequent class in the training data in both general quantity-imbalance learning (such as the popular LDAM loss[1] and Class-balanced loss[2]) and graph quantity-imbalance learning (such as GraphSMOTE[3]).
>
> [1] Learning Imbalanced Datasets with Label-Distribution-Aware Margin Loss.
>
> [2] Class-Balanced Loss Based on Effective Number of Samples.
>
> [3] GraphSMOTE: Imbalanced Node Classification on Graphs with Graph Neural Networks
>
> \
> [Q4] Response to Experiment Setting in Table 1 (lack imbalanced datasets and baselines)
>
> (Disambiguation) I assume the 'balanced/imbalanced' in your question means quantity- balanced/imbalanced.
>
> (For datasets) Table 1 and the corresponding Section 3.2 are to prove the effectiveness of our method in the pure topology imbalance scenario. According to the analysis in Section 1, topology imbalance is a ubiquitous issue in graph datasets and there is almost no dataset that is completely topologically balanced; therefore, it is more representative to choose these widely-used benchmark datasets (quantity-balanced, topology-imbalanced): although these datasets are quantity-balanced, they can still be topology-imbalanced and are good controls for studying the topology-imbalanced issue alone.
>
> (For Baselines) Topology imbalance is orthogonal to quantity imbalance: even if the training set is quantity balanced, it still can be topology imbalanced. Therefore, the existing quantity-imbalance methods are not suitable for the topology-imbalance scene. Besides, there are no existing baseline methods proposed for the topology imbalance issue according to our survey. The GCN and other models in Table1 are used to verify the generalizability of the ReNode method in different GNN architectures.

---

> > ### Comment · Reviewer_1Cph · 2021-08-31
> > **Thank you for the responses**
> >
> > I appreciate the detailed responses from the authors. However, I'm still concerned about the limited novelty of the proposed method, as well as the lack of theoretical insights. That said, I'm willing to slightly increase my score, as I believe the quality of this paper is still below the acceptance threshold.

---

> > > ### Author Response · Authors · 2021-09-01
> > > **Thank you for the further feedback**
> > >
> > > We appreciate your feedback on our comment and the decision of improving the rate.
> > >
> > > Your further concern focuses on the method. What we want to clarify is that the main purpose of this work is to reveal the **topology imbalance** issue to the community, which is widespread and harmful in the graph scene, but has not been studied so far.  Besides, the proposal of topology imbalance has been recognized by the other reviewers.
> > >
> > > The task of solving topology imbalance is non-trivial due to the asymmetry and complexity of the graph topology. All the existing methods for imbalance learning focus on quantity imbalance and can not be transferred to the topology imbalance scene.  In this work, we creatively devise the conflict detection-based solution to measure and solve the challenging topology imbalance issue.
> > >
> > > Moreover, the proposed ReNode method is lightweight and model-agnostic, which can effectively relieve topology imbalance across wide graph scenes (multiple graph datasets, various GNN models, and different imbalance settings) with limited extra computing cost. We also hope it can serve as a reliable reference for future studies on topology imbalance.

---

### Official Review · Reviewer_9Atj · 2021-07-22

**Rating:** 7
**Confidence:** 3

**Summary:**

 The issue of imbalance is a very critical problem in ML, especially in a realistic setting. In general, imbalance is quantitative, but this paper points out topological imbalance as a graph-specific problem. As a solution to topological imbalance, a method of reweighting called ReNode is proposed and its effectiveness is verified.

**Limitations And Societal Impact:**

I do not believe that this paper will have any negative impact on society.

**Main Review:**

 The problem of imbalance is a very real and important issue. The problem of imbalance for graphs has also been studied extensively in recent years. When we talk about imbalance issues, most of us consider quantitative imbalance, but the authors point out the problem of toplological imbalance. This is a very thought-provoking point, and I think it may become an important topic in the future. The authors state that this is a problem specific to graphs, but there may be similar issues with general data in the form of sampling imbalance. The idea of ReNode is relatively simple and the validation is experimental, but I think the results shown are good enough. I think ReNode is affected by the quantitative imbalance and the position of the labeled nodes, but are there any cases where ReNode might make things worse?  It seems to me that if we can see the bounds that the model shows during training, we will be able to adjust the weights better. Is it possible to adjust the weighting during training?

**Time Spent Reviewing:**

7 hours

---

> ### Author Response · Authors · 2021-08-08
> **Response to Official Review of Paper164 by Reviewer 9Atj**
>
> Thanks for your valuable feedback! Here is our response to the concerned questions.
>
> [Q1] Response to "are there any cases where ReNode might make things worse"
>
> Since it is quite difficult to directly measure the imbalance of topological distribution among different classes, we adopt the conflict detection method to indirectly reflect it; as shown in Section 4.2, our method may have a negative impact on GNN performance when the graph connectivity is poor or the labeling ratio is extremely low, in which cases the conflict level is low and topology imbalance is insufficiently reflected. Appropriate methods to solve the topology imbalance issue in these scenarios are expected in future studies.
>
> \
> [Q2] Response to "Is it possible to adjust the weighting during training"
>
> It is quite an interesting idea to dynamically adjust the training weights of nodes according to the decision boundary, which can make the adjustment more flexible and accurate. **We think it is possible** and we will study some key challenges including how to measure the bounds shift in the training process and how to design a dynamic feedback strategy to achieve this idea in the next research plan.

---

> > ### Comment · Reviewer_9Atj · 2021-08-16
> > **Comment**
> >
> > Thanks for your response. I understand the current position of this research and your perception. I look forward to future developments.

---

### Decision · Program_Chairs · 2021-09-28

**Decision:**

Accept (Poster)

**Comment:**

This paper points out and investigates the "topology-imbalance" problem of node representation learning on graph-structured data. Unlike the "quantity-imbalance" problem, the topology imbalance is caused by the topological properties of the labeled nodes, i.e., the locations of the labeled nodes on the graph can influence how information is spread over the entire graph. As it involves the graph topology, the topology imbalance is much harder to quantify than the quantity imbalance. In this work, the authors demonstrate how this issue can impact the semi-supervised node classification based on graph neural networks (GNN). The authors then propose a simple proxy to measure the topology imbalance and re-weighting based method to correct for this imbalance.

As pointed out by Reviewer 9Atj and t36c, this paper introduces a novel research problem of semi-supervised node classification on graphs, which I consider as a positive aspect of this paper. On the other hand, there are a couple of criticisms on the simplicity of the proposed method and lack of theoretical guarantees (Reviewer 1Cph and t36c). As this problem is previously unknown to the community and the task of solving topology imbalance is non-trivial due to the asymmetry and complexity of the graph topology, I can understand the point that the authors raise in their response: "the main purpose of this work is to reveal the topology imbalance issue to the community". Hence, the simplicity of the proposed method stems from the need to effectively communicate to the community that the problem does exist and the proposed method can be used to solve this problem to certain extent. However, several challenges remain to be addressed to fully alleviate this problem.

Based on my own evaluation of the paper and reviews of the expert reviewers, I view the introduction of a novel research problem as a major contribution of this paper that will stimulate subsequent works in this direction. Hence, I recommend this paper for acceptance as a poster at NeurIPS2021.

**Consistency Experiment:**

NeurIPS has a long history of experimentation. In 2014, NeurIPS ran an experiment in which 10% of submissions were reviewed by two independent committees to quantify the randomness in the review process. This year, we repeated a variant of this experiment to see how the quality of the review process has changed over time.  This paper was part of the experiment and was therefore assigned to two committees (consisting of reviewers, an Area Chair, and a Senior Area Chair) that reached independent decisions.  If both committees made the same recommendation, this recommendation was followed. If a single committee recommended acceptance, the paper was accepted (with the exception of a few cases in which the other committee identified what we considered a fatal flaw, e.g., an error in a key result).

This copy’s committee reached the following decision: **Accept (Poster)**

The other committee assigned to the paper recommended **Reject**.  You can find the other set of reviews, along with any follow up discussion with the authors here:
https://openreview.net/forum?id=w3x8K0M6sAz